

**The Open-source Data Inventory for Anthropogenic Carbon dioxide (CO$_2$), version**
**2016 (ODIAC2016): A global, monthly fossil-fuel CO$_2$ gridded emission data product for**
**tracer transport simulations and surface flux inversions**
Tomohiro Oda[1,2], Shamil Maksyutov[3] and Robert J. Andres[4]
1: Global Modeling and Assimilation Office, NASA Goddard Space Flight Center, Greenbelt,
MD, USA.
2: Goddard Earth Sciences Technology and Research, Universities Space Research
Association, Columbia, MD, USA
3: Center for Global Environmental Research, National Institute for Environmental Studies,
Tsukuba, Ibaraki, Japan
4: Carbon Dioxide Information Analysis Center, Oak Ridge National Laboratory, Oak Ridge,
TN, USA
Corresponding author: T. Oda (tomohiro.oda@nasa.gov)
**Abstract**
Open-source Data Inventory for Anthropogenic CO$_2$ (ODIAC) is a global high-spatial
resolution gridded emission data product that distributes carbon dioxide (CO$_2$) emissions
from fossil fuel combustion. The emission spatial distributions are estimated at a 1×1 km
spatial resolution over land using power plant profiles (emission intensity and geographical
location) and satellite-observed nighttime lights. This paper describes the latest version of the
ODIAC emission data product (ODIAC2016) and presents analyses that help guiding data
users, especially for atmospheric CO$_2$ tracer transport simulations and flux inversion analysis.
Since the original publication in 2011, we have made modifications to our emission modeling
framework in order to deliver a comprehensive global gridded emission data product. Major
changes from the 2011 publication are 1) the use of emissions estimates made by the Carbon
Dioxide Information Analysis Center (CDIAC) at Oak Ridge National Laboratory (ORNL)
by fuel type (solid, liquid, gas, cement manufacturing, gas flaring and international aviation
and marine bunkers), 2) the use of multiple spatial emission proxies by fuel type such as
nightlight data specific to gas flaring and ship/aircraft fleet tracks and 3) the inclusion of
emission temporal variations. Using global fuel consumption data, we extrapolated the
CDIAC emissions for the recent years and produced the ODIAC2016 emission data product
that covers 2000-2015. Our emission data can be viewed as an extended version of CDIAC
gridded emission data product, which should allow data users to impose global fossil fuel
emissions in more comprehensive manner than original CDIAC product. Our new emission
modeling framework allows us to produce future versions of ODIAC emission data product
with a timely update. Such capability has become more significant given the CDIAC's
shutdown. ODIAC data product could play an important role to support carbon cycle science,
especially modeling studies with space-based CO$_2$ data collected near real time by ongoing
carbon observing missions such as Japanese Greenhouse Observing SATellite (GOSAT),
NASA's Orbiting Carbon Observatory 2 (OCO-2) and upcoming future missions. The
ODIAC emission data product is distributed from http://db.cger.nies.go.jp/dataset/ODIAC/
with a DOI.



## 1. Introduction

Carbon dioxide ($CO_2$) emissions from fossil fuel combustion are the main cause for the observed increase in atmospheric $CO_2$ concentration. The Carbon Dioxide Information Analysis Center (CDIAC) at Oak Ridge National Laboratory (ORNL) estimated that the global total fossil fuel $CO_2$ emissions (FFCO2; fuel combustion, cement production and gas flaring) in the year 2014 was 9.855 PgC based on fuel statistics data published by United Nation (U.N.) (Boden et al., 2017). This FFCO2 estimate often serves as a reference in carbon budget analysis, especially for inferring $CO_2$ uptake by terrestrial biosphere and oceans (e.g. Ballantyne et al., 2012; Le Quéré et al., 2016). The Global Carbon Project for example estimated that approximately 55% of the carbon released to the atmosphere (FFCO2 plus emissions from land use change) was taken up by natural sinks over the past decade (2006-2015) (Le Quéré et al., 2016).

Similarly, FFCO2 estimates serve as a reference in atmospheric $CO_2$ flux inversion analysis where the location and size of natural sources and sinks are estimated using atmospheric $CO_2$ data and atmospheric transport models (e.g. Tans et al., 1990; Bousquet et al., 1999; Gurney et al., 2002; Baker et al., 2006). In the conventional inversion method, unlike land and oceanic fluxes, FFCO2 is a given quantity and never optimized (e.g. Gurney et al., 2005). FFCO2 thus needs to be accurately quantified and given in space and time to yield robust estimates of natural fluxes (Gurney et al., 2005). Accurately prescribing FFCO2 has become more critical because of the use of spatially and temporally dense $CO_2$ data from a wide variety of observational platforms (ground-based, aircrafts and satellites), which inform not only background levels of $CO_2$ concentration, but also $CO_2$ contributions from anthropogenic sources (e.g. Schneising et al., 2013; Janardanan et al., 2016; Hakkarainen et al., 2016). Atmospheric transport models then need to be run at a higher spatiotemporal resolution than before to fully interpret and utilized $CO_2$ variability observed at synoptic to local scale to quantify sources and sinks (e.g. Feng et al. 2016; Lauvaux et al., 2016). FFCO2 data thus needs to be accurately given at a high resolution so as not to cause biases in simulations.

Global FFCO2 data are available in a gridded form from different institutions and research groups (e.g. CDIAC/ORNL and Europe's Joint Research Center (JRC)) and those gridded emission data are often based on disaggregation of national (or sectoral) emissions (e.g. Andres et al., 1996; Rayner et al., 2010; Oda and Maksyutov 2011; Janssens-Maenhout et al., 2012; Kurokawa et al., 2013; Asefi-Najafabady et al., 2014). The emission spatial distributions are often estimated using spatial proxy data that approximate the location and intensity of human activities (hence, $CO_2$ emissions) (e.g. population, nighttime lights and gross domestic production (GDP)) and/or geolocation of specific emission sources (e.g. power plant, transportation, cement production/industrial facilities and gas flares). CDIAC gridded emission data product for example is based on an emission disaggregation using population density at a 1×1 degree resolution (Andres et al., 1996). The Emission Database for Global Atmospheric Research (EDGAR, http://edgar.jrc.ec.europa.eu/) estimates emissions on the emission sectors specified by the Intergovernmental Panel on Climate Change (IPCC) methodology instead of fuel type and use spatial proxy data and geospatial data such as point and line source location at a 0.1×0.1 degrees (Janssens-Maenhout et al., 2012).

Satellite-observed nighttime light data has been identified as an excellent spatial indicator for human settlements and intensities of some specific human activities (e.g. Elvidge et al., 1999, 2009) and has been used to infer the associated $CO_2$ emissions or their spatial distributions (e.g. Doll et al., 2000, Ghosh et al., 2010, Rayner et al., 2010). Oda and



Maksyutov (2011) proposed a combined use of power plant profiles (power plant emission
intensity and geographical location) and nighttime light data to achieve a global high-spatial
resolution emission field. The decoupling of the point source emission which often have less
spatial correlation with population (hence, nighttime light), yields an improved high-
resolution emission field that shows an improved agreement with the U.S. 10km Vulcan
emission product developed by Gurney et al. (2009) (Oda and Maksyutov 2011). Based on
Oda and Maksyutov (2011), we initiated a high-resolution emission data development (named
as the Open-source Data Inventory for Anthropogenic $CO_2$, ODIAC) under the Japanese
Greenhouse Gases Observing SATellite (GOSAT, Yokota et al., 2009) at the Japanese
National Institute for Environmental Studies (NIES). The original purpose of the emission
data development was to provide an accurate prior FFCO2 field for global and regional $CO_2$
inversions using the column-averaged $CO_2$ ($X_{CO2}$) data collected by GOSAT. Since 2009, the
ODIAC emission data product has been used for the inversion for the official GOSAT Level
4 (surface $CO_2$ flux) data production (Takagi et al., 2009; Maksyutov et al., 2013), NOAA's
CarbonTracker (Peters et al., 2007) as a supplementary FFCO2 data, as well as dozens of
published works (e.g. Saeki et al., 2013; Thompson et al., 2015; Feng et al., 2016; Feng et al.,
2017; Shirai et al., 2017) including several urban scale modeling studies (e.g. Ganshin et al.
2010; Oda et al., 2012; Brioude et al., 2013; Lauvaux et al., 2016; Janardanan et al., 2016;
Oda et al., 2017).
In response to increasing needs from the $CO_2$ modeling research community, we have
upgraded and modified our modeling framework in order to produce a global, comprehensive
emission data product on timely manner, while our flagship high-resolution emission
modeling approach remains as the same. In this manuscript, we describe the latest version of
the ODIAC emission data product (ODIAC2016, 2000-2015) along with the emission
modeling framework we are currently based on, highlighting changes/differences from Oda
and Maksyutov (2011).
**2. Emission modeling framework**
Fig. 1 illustrates our current ODIAC emission modeling framework (we defined it as
"ODIAC 3.0 model", in contrast to the original version). Major changes/differences from Oda
and Maksyutov (2011, ODIAC v1.7) are 1) the use of emissions estimates made by the
CDIAC (rather than our own emission estimates), 2) the use of multiple spatial emission
proxies in order to distribute CDIAC emissions made by fuel type, and 3) the inclusion of
emission temporal variations (version 1.7 only indicates annual emission fields). Given
CDIAC estimates have been one of well-respected, widely-used in the carbon research
community (e.g. Ballantyne et al., 2012; Le Quéré et al., 2016), our philosophy in our
emission data development is we develop and deliver an extended, comprehensive global
gridded emission data product, fully utilizing CDIAC emissions data (e.g. emission estimates
in both tabular and gridded forms). We also extend/upgrade CDIAC emission data where
possible. Our emission modeling framework was also designed to produce an emission data
product in a timely manner, with updated information. As our ODIAC data product is based
CDIAC emission data, our emission data production capability is significant given the
expected discontinuity of future CDIAC emission data.
Starting with national emission estimates as an input, our model framework achieves
monthly, global FFCO2 gridded fields via preprocessing, and spatial and temporal
disaggregation. CDIAC national estimates made by fuel type (liquid, gas, solid, cement



production, gas flare and international bunker emissions) are further divided into an extended
set of ODIAC emission categories (point source, non-point source, cement production, gas
flare, international aviation and marine bunker (further described in section 3). It is important
to note that ODIAC2016 carries emissions from international bunker (international marine
bunker and aviation, often accounts for few percent of the global total emissions), which are
not included in the CDIAC gridded emission data products (CDIAC gridded emission data
only indicate national emissions and international bunker emissions are often not considered
to be a part of national emissions in an international convention). With the inclusion of
international bunker emissions, we provide more comprehensive global gridded emission
field. We extended CDIAC estimates over the recent years that was not yet covered in the
version of CDIAC estimates (2014-2016), in order to support near-real time $CO_2$
simulations/analysis. Emissions are then spatially distributed using a wide variety of spatial
data (e.g. point source geographical location, nighttime light data and flight/ship tracks,
further described in section 4). We adopts emission seasonality from existing emission
inventories for particular emission categories (further described in section 5).
In the following sections (section 3-5), we describe how ODIAC2016 was developed. It is
important to note that ODIAC2016 is based on the best available data at the time of the
development (ODIAC2016 was released in September 2016). Thus, some of the emission
estimates and underlying data used in ODIAC2016 might have been outdated. For traceability
purpose, data used in this development, their versions/editions, and data sources are
summarized in Appendix A. Following the results and evaluation section (section 6), we
discuss caveats and current limitations in our modeling framework/emission data product
(section 7), and then describe how we will update ODIAC emission data product with
updated fuel statistics and/or emission information (section 8). Given recent most of
atmospheric $CO_2$ inversion studies focused on years after 2000, we put a priority to develop
emission data for years after 2000 and deliver to the science community in a timely manner.
Future versions of ODIAC data however might have a longer, extended time coverage.
Currently ODIAC data are provided in two data formats: 1) global 1×1 km (30 arc second)
monthly data in GeoTIFF format (only includes emissions over land) and 2) 1×1 degree
annual (12 month) data in netCDF format (includes international bunker emissions). The
improvements with the use of improved nighttime light data in the 1×1 km data were
documented in Oda et al. (2012). This manuscript thus focuses on the comprehensive global
FFCO2 fields at a 1×1 degree, otherwise specified.


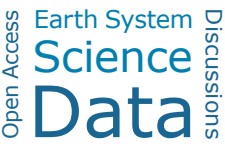

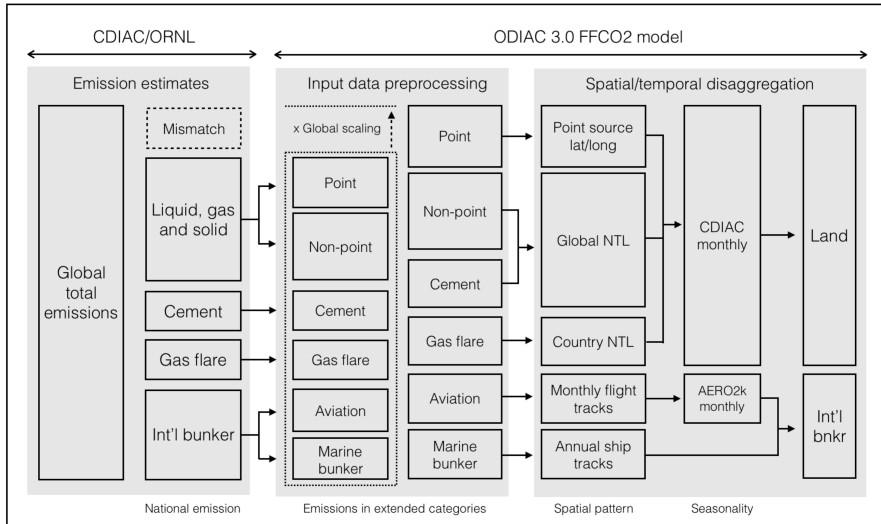

**Figure 1.** A schematic figure of the ODIAC emission modeling framework (defined as "ODAIC 3.0 FFCO2 model"). Starting with CDIAC national emission estimates made by fuel type (emission estimates), the CDIAC emission estimates are first divided into extended ODIAC emission categories (input data processing, see section 3). ODIAC 3.0 FFCO2 model then distributes the emissions in space and time, using point source geolocation information and spatial data depending on emission category such as nighttime light (NTL), and aircraft and ship fleet tracks (spatial disaggregation, see section 4). The emission seasonality for emissions over land and international aviation were adopted from existing emission inventories (temporal disaggregation, see section 5).

**3. Emission estimates and input emission data preprocessing**
3.1 Emissions for 2000-2013
CDIAC FFCO2 emissions estimates are based on fuel statistic data published as United
Nation Energy Statistics Database (Boden et al., 2017). Emission estimates are calculated on
global, national and regional basis and by fuel type in the method described in Marland and
Rotty (1984). CDIAC also provides their own gridded emission data products that indicate
annual and monthly FFCO2 fields at a 1×1 degree (Andres et al., 1996; Andres et al., 2011).
ODIAC2016 is primarily based on the year 2016 version of the CDIAC national estimates
(Boden et al., 2016), which was the most up-to-date CDIAC emission estimates at the time of
the data development (currently Boden et al. 2017 is the latest). We first aggregated the
CDIAC national (and regional) emissions estimates to 65 countries and 6 geographical
regions (North America, South and Central Americas, Europe and Eurasia, the Middle East,
Africa, and Asia Pacific) defined in Oda and Maksyutov (2011) (see the country/region
definitions are shown in Table 1 in Oda and Maksyutov 2011). In addition to the national and
geographical categories, we decided to include Antarctic fishery emissions, which are from
fishery activities over the Antarctic Ocean (< 60S, 1~ 4 kTC/yr over 1987-2007 by Boden *et*
*al.,* 2016), as an individual emission region and distributed in the same way as Andres et al.



(1996). Emissions from international bunker and aviation are not included in national emissions by international convention. Thus CDIAC gridded emission data products do not include the emissions from international bunker and aviation although CDIAC do have records of those emissions on national/regional basis. ODIAC2016 includes those emissions to achieve comprehensive global FFCO2 gridded emission fields.

In CDIAC emission estimates, the global total emission and national total emissions are obtained by different calculation methods (global fuel production vs. apparent national fuel consumption, see Andres et al., 2012) and the CDIAC national totals do not sum to the CDIAC global total due to the difference in calculation method and inconsistencies in the underlying statistical data (e.g. import/export totals) (e.g. Andres et al., 2012). We thus calculate the difference between the global total and the sum of national totals and scaled up national totals to account for the difference. Andres et al. (2014) report global total emission estimates calculated with production data (as opposed to apparent consumption data) have the smallest uncertainty (approximately 8% (2 sigma). It is thus used as the reference for global carbon budget analysis (e.g. Le Quéré et al., 2016). Inversion analysis is an extended version of the global carbon budget analysis using atmospheric models. We thus believe that imposing transport models and/or inversion models in a consistent way with the global carbo budget analysis such as Le Quéré et al. (2016) has significance, although we sacrifice the accuracy of the national/regional emission estimates. Due to the global scaling, national totals in ODIAC2016 differ from the estimates originally reported by CDIAC. The differences between the CDIAC global total and the sum of national emissions are often few percent and thus the magnitude of the scaling is often within the uncertainty range of national emissions (e.g. 4.0 to 20.2%, Andres et al., 2014).

3.2 Emissions for 2014-2015

The year 2016 version of the CDIAC estimates only covers years to 2013 (Boden et al., 2016). We thus extrapolated the year 2013 CDIAC emissions to years 2014 and 2015 using the year 2016 version of BP global fuel statistical data (BP, 2017). Our emission extrapolation approach are the same as Myhre et al. (2009) and Le Quéré et al. (2016). Emissions from cement production and gas flaring (approximately 5.7% and 0.6% of the 2013 global total, Boden et al., 2016) were assumed to be as the same as year 2013. International bunker emissions were scaled using changes in national total emissions.

3.3 CDIAC emission sector to ODIAC emission categories

CDIAC national emission estimates (prepared by fuel type) were re-categorized to our own ODIAC emission categories (point source, nonpoint source, cement production, gas flare and international aviation and international marine bunker). Following Oda and Maksyutov (2011), the sum of emissions from liquid, gas and solid fuels was further divided into point source emissions and non-point source emissions. The total emissions from point sources were estimated using national total power plant emissions calculated using CARMA (Oda and Maksyutov, 2011). As mentioned earlier, CDIAC gridded emission data products only indicate national emissions and do not include international bunker emissions (Andres et al., 1996, Andres et al., 2011). In contrast, EDGAR provides bunker emissions in their gridded data product (JRC, 2017). Peylin et al. (2013) show some models include international bunker emissions and some do not, although the difference due to the inclusion/exclusion of the international bunker emissions in the prescribed emissions could be corrected afterwards



(Peylin et al., 2013). In ODIAC2016, we carry CDIAC international bunker emissions
reported on country basis to achieve the complete picture of the global fossil fuel emissions.
Country total bunker emissions (aviation plus marine bunker) were distributed using spatial
proxy data adopted from other emission inventories described later (see section 4.3).
Although CDIAC does not report emissions from international aviation and marine bunker
separately, we loosely estimated those two emissions using U.N. statistics. We estimated the
fraction of aircraft emissions using jet fuel and aviation gasoline consumption and then the
international bunker emissions were divided into aircraft and marine bunker emissions.
**4. Spatial emission disaggregation**
4.1 Emissions from point sources, non-point sources and cement production
We define the sum of the emissions from solid, liquid and gas fuels as land emission (see
Fig. 1). Land emissions are further divided into two emission categories (point source
emissions and non-point source emissions) and then distributed in the ways described in Oda
and Maksyutov (2011): Point source emissions are mapped using power plant profiles
(emission intensity and geographical location) and non-point source emissions are distributed
using nighttime light data collected by the Defense Meteorological Satellite Program (DMSP)
satellites. To avoid a difficulty in emission disaggregation especially over bright regions in
nighttime light data (e.g. cities), Oda and Maksyutov (2011) employed a product that does not
have an instrument saturation issue, rather than regular nightlight product. ODIAC2016
employs the latest version of the special nighttime light product (Ziskin et al., 2010). The
improved nighttime light data has mitigated the underestimation of emissions over dimmer
areas seen in ODIAC v1.7 (Oda et al., 2010). Nighttime light data are currently available for
multiple years (1996-97, 1999, 2000, 2002-03, 2004, 2005-06 and 2010). In ODIAC2016,
due to the lack of information, emissions from cement production were spatially distributed as
a part of non-point source emissions, although those emissions should have been distributed
as point sources. This needs to be fixed in future versions in ODIAC emission data.
4.2 Emissions from gas flaring
In the ODIAC v1.7, emissions from gas flaring were not considered (Oda and Maksyutov
2011). Nighttime light pixels corresponding to gas flares often appear very bright and would
result in creating strong point sources in emission data (Oda and Maksyutov, 2011). We thus
identified and excluded those bright gas flare pixels before distributing land emissions, using
another global nighttime light data product that was specifically developed for gas flares by
National Oceanic and Atmospheric Administration (NOAA), National Centers for
Environmental Information (NCEI, former National Geophysical Data Center (NGDC)) (Oda
and Maksyutov, 2011). In ODIAC2016 we separately distributed CDIAC gas flare emissions
using the $1 \times 1$ km nightlight-based gas flare maps developed for 65 individual countries
(Elvidge et al., 2009). Other than the 65 countries, gas flare emissions were distributed as a
part of land emissions.
4.3 Emissions from international aviation and marine bunker





Emissions from international aviation and marine bunker were distributed using aircraft and
ship fleet tracks. International aviation emissions were distributed using the AERO2k
inventory (Eyers et al., 2005). The AERO2k inventory was developed by a team at
Manchester Metropolitan University (MMU) and indicates fuel use and $NO_x$, $CO_2$, CO,
hydrocarbon and particulate emissions for 2002 and 2025 (projected) with injection height at
a $1 \times 1$ degree spatial resolution on monthly basis. We used their column total $CO_2$ emissions
to distribute emissions to a single layer. International marine bunker emissions were
distributed at a $0.1 \times 0.1$ degree using an international marine bunker emission map from the
EDGAR v4.1(JRC, 2017). We decided not to adopt an international and domestic shipping
(1A3d) map from the EDGAR v4.2 as it includes domestic shipping emissions that we does
not distinguish.
**5. Temporal emission disaggregation**
The inclusion of the temporal variations is often a key in transport model simulation. For
$CO_2$ flux inversion, the potential biases in flux inverse emission estimates due to the lack of
temporal profiles was suggested by Gurney et al. (2005). In ODIAC2016, we adopt the
seasonal emission changes developed by Andres et al. (2011). The CDIAC monthly gridded
data include monthly national emissions gridded at a $1 \times 1$ degree resolution (Andres et al.
2011). We normalized the monthly emission fields by the annual total and the applied to our
annual emissions over land. The seasonality in ODIAC2016 is based on the year 2013 version
of the CDIAC monthly gridded emission. The CDIAC monthly emission data do not cover
the recent years. For recent years, we created a climatological seasonality using monthly
CDIAC data from 2000-2010 (excepting 2009 where economic recession happened). Due to
the limited availability of monthly fuel statistical data, Andres et al. (2011) used proxy
country and also seasonality allocated by Monte Carlo simulations. The years between 2000-
2010 were most data rich period and mostly explained by data (see Fig. 1 in Andres et al.,
29 2011).
Although ODIAC2016 only provides monthly emission fields, users can derive hourly
emissions by applying scaling factors developed by Nassar et al. (2013). The Temporal
Improvements for Modeling Emissions by Scaling (TIMES) is a set of scaling factors which
one can derive weekly emissions and diurnal emissions from any monthly emission data that
you use. Temporal profiles are collected from Vulcan, EDGAR and best available
information and gridded on a $0.25 \times 0.25$ degree (Nassar et al., 2013). TIMES also includes
per capita emissions corrections for Canada (Nassar et al., 2013).

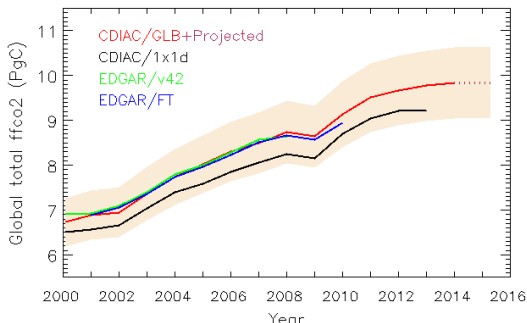

**Figure 2.** Global emission time series from four gridded emission data: CDIAC (red, 2000-2013) plus projected emissions (dashed maroon, 2014-2015) (values taken from ODIAC), CDIAC 1×1 degree (black, 2000-2013), EDGAR v4.2 (green, 2000-2008) and EDGAR v4.2 Fast Track (blue, 2000-2010). The values here are given in the unit of peta gram (= giga tonnes) carbon per year. The shaded area indicated in tan is a two-sigma uncertainty range (8%) estimated for CDIAC global total emission estimates by Andres et al. (2014).

**6. Results and discussions**
6.1 Annual global emissions
In Fig. 2, global emission time series from different emission data were compared to give an
idea of agreement among them. We calculated the global total for each year from four gridded
emission data for the period of 2000-2016: CDIAC global total + projection (taken from
ODIAC2016), CDIAC gridded data (hence, no international bunker emissions), two versions
of EDGAR gridded data (v4.2 and FastTrack). The uncertainty range (shaded in tan) is 8% (2
sigma) estimated for CDIAC global by Andres et al. (2014). Those gridded emission data are
often used in global atmospheric $CO_2$ inversion analysis (e.g. Peylin et al., 2013). To account
for the difference in emission reporting categories (e.g. fuel basis in CDIAC vs. emission
sector basis in EDGAR), the EDGAR totals were calculated as the "total short cycle C"
emissions minus the sum of emissions from agriculture (IPCC code: 4C and 4D), land use
change and forestry (5A, C, D, F and 4E) and waste (6C) (see more details on emission
sectors documented in JRC (2017)). International aviation (1A3a) and navigation (1A3b)
were thus included in values for EDGAR time series. The authors acknowledge the JRC has
updated EDGAR emission time series for 1970-2012 in November 2014 (JRC, 2017). This
study however uses gridded emission data, which are not fully based on the updated emission
estimates, in order to characterize differences from gridded emission data, especially for
potential data users in the modeling community.
All four global total values obtained from four gridded emission data agree well within 8%
uncertainty. The difference between ODIAC and CDIAC (3.3%-5.7%) were largely
attributable to the international bunker emissions and global correction. ODIAC (where the
total was scaled by CDIAC global total) and two versions EDGAR showed minor differences
in magnitude (0.3%-2.7%) and trend, which are largely attributable to the differences in the





underlying statistical data (e.g. U.N. Stat vs. EIA from different inventory years) and the
emission calculation method (fuel basis vs. sector basis).  Global total estimates at 5-year
increments are shown in Table 1.  For the year 2014 and 2015, we estimated the global total
emissions 9.836 and 9.844 PgC. Boden et al. (2017) reported the latest estimate for year 2014
global total emission as 9.855 PgC. Our projected 2014 emission estimate was lower than the
latest estimate by approximately 0.02 PgC (0.2%).
**Table 1**. Global total emission estimates for year 2000, 2005 and 2010 from four gridded
emission data (ODIAC2016, CDIAC, EDGAR v4.2 and EDGAR FastTrack). Values for two
versions of EDGAR emission data were calculated by subtracting emissions from agriculture
(IPCC code: 4C and 4D), land use change and forestry (5A, C, D, F and 4E) and waste (6C)
from the total EDGAR $CO_2$ emissions (total short cycle C).

| Year | ODIAC2016 | CDIAC national | EDGAR v4.2 | EDGAR FT |
|------|-----------|----------------|------------|----------|
| 2000 | 6727 | 6506 (-3.3%) | 6907 (+2.7%) | N/A |
| 2005 | 8025 | 7592 (-5.4%) | 8005 (-0.2%) | 7959 (-0.8%) |
| 2010 | 9137 | 8694 (-4.8%) | N/A | 8950 (-2.0%) |



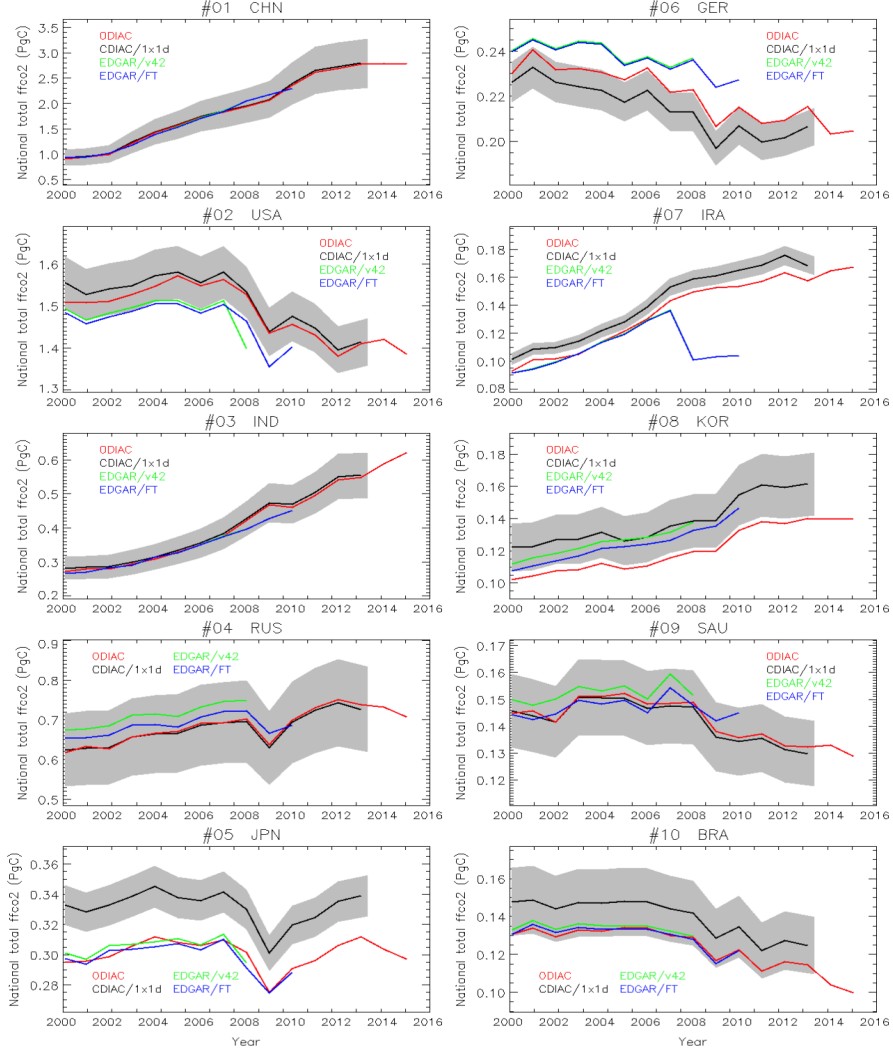

**Figure** 3. National emission time series for top 20 emitting countries (China, U.S., India, Russian Federation, Japan, Germany, Islamic Republic of Iran, Republic of Korea (South Korea), Saudi Arabia and Brazil). The values are given in the unit of peta gram (=giga tonnes) carbon per year. The values are calculated using gridded emission data, not tabular emission data. The national total values in the plots might be thus different from values indicated in the tabular form due to the emission disaggregation. The shaded area in grey indicates a two-sigma uncertainty range estimated by Andres et al. (2014) (see Table 2).

Fig. 3 shows the same type of comparison as Fig. 2, but for the top 10 emitting countries
(China, US, India, Russian Federation, Japan, Germany, Islamic Republic of Iran, Republic of
Korea (South Korea), Saudi Arabia and Brazil, according to the year 2013 ranking reported
by CDIAC). We aggregated all the four gridded emission fields to a common 1×1 degree field



and sampled using the 1×1 degree country mask used in CDIAC emission data development.
The annual uncertainty estimates for national total emissions (2 sigma) are made following
the method described by Andres *et al.,* (2014) and values are shown in Table 2. In the
analysis presented in Fig. 3, emissions from international aviation (1A3a) and navigation
(1A3b) are excluded. All four national total values sampled from four gridded emission data
at a 1×1 degree often agree within the uncertainty estimated by Andres et al. (2014).
Systematic differences of ODIAC from CDIAC can be largely explained by 1) global
correction (the total was scaled using CDIAC global total) and 2) the differences in emissions
disaggregation methods. Although ODIAC is expected to indicate slightly higher values than
CDIAC (often few percent) because of the global correction (note global correction can be
negative, despite of the depiction in Fig. 1), ODIAC sometimes indicates values lower that
CDIAC more than few percent (see Japan in Fig. 3 as an example). This is due to a sampling
error using the 1×1 degree country map in the analysis. The aggregated 1×1 degree ODIAC
field is slightly larger than that of CDIAC especially because of the coastal areas depicted a
high-resolution in the original 1×1 km emission field. This type of sampling error was
discussed in Zhang et al. (2014). ODIAC employs a 1×1 km coastline and a 5×5 km country
mask as described in Oda and Maksyutov (2011). Thus, the use of 1×1 degree CDIAC
country map results in missing some land mass (hence, $CO_2$ emissions). Similar sampling
error can happen for countries that are physical small and island countries, depending on the
resolution of analysis. Despite of the sampling error, the authors used the CDIAC 1×1 degree
country map to do this comparison analysis with having CDIAC as a reference. The lower
emission indicated by ODIAC or EDGAR in this analysis does not always mean the national
total emissions are lower. The emission estimates at national level often agree well even
among different emission inventories (e.g. Andres et al., 2012).
**Table 2**. Annual uncertainty estimates associated with CDIAC national emission estimates.
The uncertainty estimates were made following the method described by Andres et al. (2014).
The national total emissions for the year 2013 were taken from Boden et al. (2016).

| Ranking # | Country | 2013 emissions in kTC (% of the global total) | Uncertainty (%) |
|---|---|---|---|
| 1 | China | 2,795,054 (28.6%) | 17.5 |
| 2 | U.S. | 1,414,281 (14.5%) | 4.0 |
| 3 | India | 554,882 (5.7%) | 12.1 |
| 4 | Russia Federation | 487,885 (5.0%) | 14.8 |
| 5 | Japan | 339,074 (3.5%) | 4.0 |
| 6 | Germany | 206,521 (2.1%) | 4.0 |
| 7 | Islamic Republic of Iran | 168,251 (1.7%) | 9.4 |
| 8 | Republic of Korea | 161,576 (1.7%) | 12.1 |
| 9 | Saudi Arabia | 147,649 (1.5%) | 9.4 |
| 10 | Brazil | 137,354 (1.4%) | 12.1 |




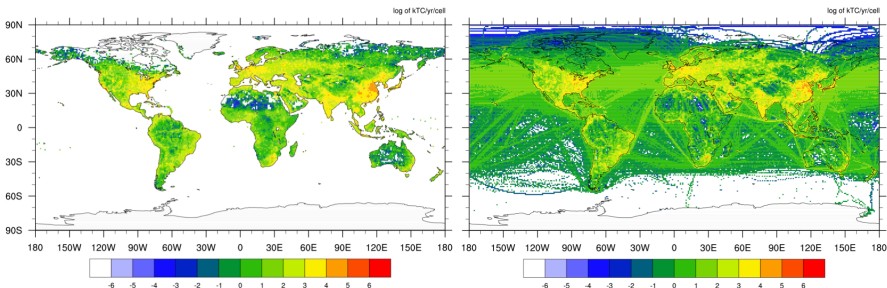

**Figure 4**. Year 2013 global fossil fuel $CO_2$ emissions distributions from CDIAC (left, 8.36 PgC) and ODIAC (right, 9.78 PgC). The ODIAC emission field was aggregated to a common $1 \times 1$ degree resolution. The value is given in the unit of log of thousand tonnes C/cell.

6.2 Global emission spatial distributions
5        The global total emission fields of CDIAC gridded emission data and ODIAC2016 for the
year 2013 (the most recent year CDIAC indicates) are shown in Fig. 4. Emission fields are
shown at a common 1×1 degree. The major difference seen between two fields is primarily
due to inclusion/exclusion of emissions from international bunker emissions that largely
account for the differences indicated in Table 1. A breakdown of ODIAC year 2013 emission
field are presented by emission category in Fig. 5. Emission fields for point sources, non-
point sources, cement production and gas flaring were produced at a 1×1 km resolution in
ODIAC 3.0 model, but as mentioned earlier, we focus on the 1×1 degree version of
ODIAC2016 in this manuscript. In CDIAC gridded emission data, those emissions are
distributed by population data without fuel type distinction. In ODIAC 3.0 model, we have
added additional layers of consideration in the emission modeling from the conventional
CDIAC model and add the possibility of future improvement with improved emission proxy
data.





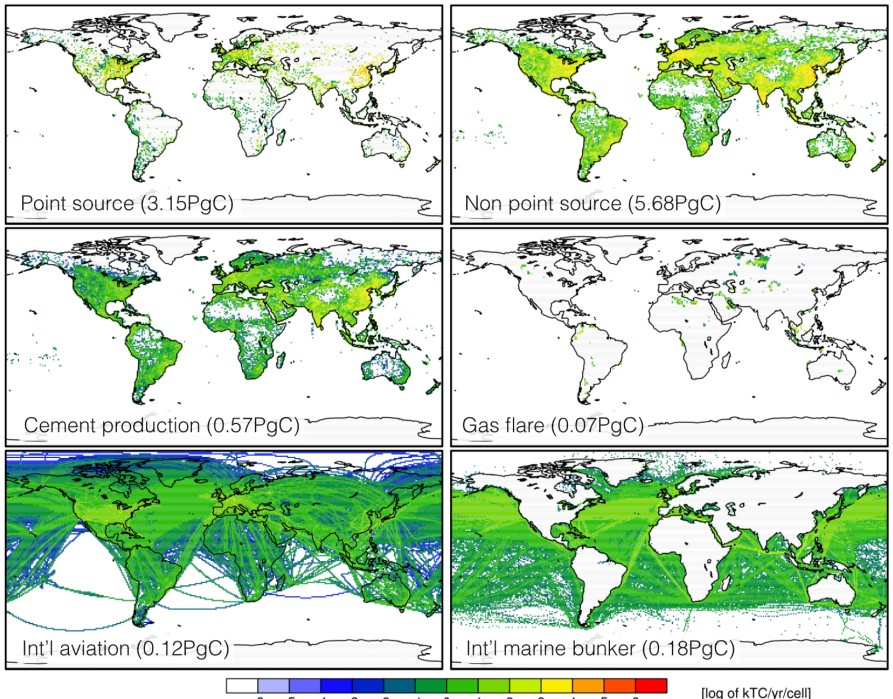

**Figure 5**. Year 2013 global distributions of ODIAC fossil fuel emissions by emission type. The panels show emissions from (from top to the right, then down) point source, non-point source, cement production, gas flaring, international aviation and international shipping. The values in the figures are given in the unit of log of thousand tonnes carbon/year/cell (1 × 1 degree). The numbers in the brackets are the total for the category emissions in the unit of PgC (total year 2013 emission in ODIAC2016 was 9.78 PgC).

In Fig. 6, we compared the four global gridded products over land and also calculated
differences from ODIAC2016 (shown in Fig. 7). It is often very challenging to evaluate the
accuracy and uncertainty of gridded emission data, because of the lack of direct physical
measurements at grid scales (Andres et al., 2016). Recent studies have attempted to evaluate
the uncertainty of gridded emission data by comparing emission data each other (e.g. Oda et
al., 2015; Hutchins et al., 2016). The differences among emission were used as a proxy for
uncertainty. However, it is important note that such evaluation does not give us an objective
measure of which one is closer to truth, beyond characterizing the differences in emission
spatial patterns and magnitudes from methodological viewpoints (e.g. emission estimation
and disaggregation). Some of the gridded emission data are partially disaggregated using
commercial information, which users are often not authorized to fully disclose the
information used and thus makes the comparison even less meaningful and/or significant.
Oda et al. (2015) also discussed that emission inter-comparison approaches often do not allow
us to evaluate two distinct uncertainty sources (emissions and disaggregation) separately. In
addition, because of the use of emission proxy for emission disaggregation (rather than
mechanistic modeling), such comparison can be only implemented at an aggregated, coarse
spatial resolution. These issues will be further discussed in the Section 7.



Because of the limitation mentioned above, we here compared emission data only to
characterize the differences that can be explained by the differences in emission
disaggregation methods. We implemented this comparison exercise using 2008 emission field
aggregated at a 1×1 degree resolution. Year 2008 is the most recent year where all the four
emission fields are available. The major emission spatial patterns (e.g. emitting regions such
as North America, Europe and East Asia) are overall very similar as the correlations were
driven by national emission estimates (which we already saw good agreement earlier), but we
do see differences due to emission disaggregation at subnational scale. Because of the use of
nightlight, ODIAC did not indicate emissions over some of the areas (e.g. Africa and Eurasia)
while others do. Especially, EDGAR has emissions over those areas that are largely explained
by line source emissions such as transportation. Overall, ODIAC tends to put more emissions
towards populated areas than suburbs. This is also explained by the lack of line sources. In
EDGAR v4.2, domestic fishery emissions can be seen, but not in EDGAR FT. Even in these
two EDGAR versions, we can confirm the subnational differences at United States, Europe
and China.

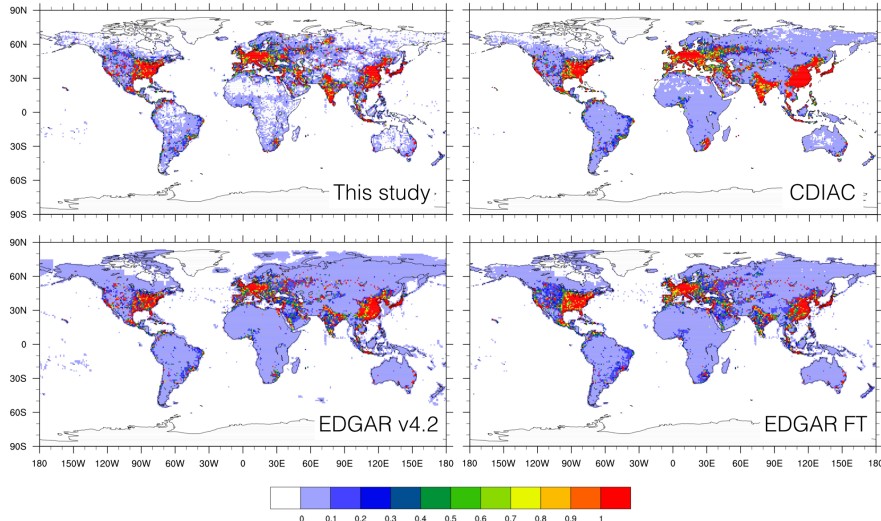

**Figure 6**. Land emissions from ODIAC (upper left), CDIAC (upper right), two versions of
EDGAR emission data (v4.2 lower left and v4.2 Fast Track lower right). The units are tonnes
carbon/year/ cell (1×1 degree). In addition to excluding emissions from international aviation
and marine bunker, some of the sector emissions were subtracted from EDGAR short cycle
total emissions to account for the differences in emission calculation methods between
CDIAC and EDGAR, as also done earlier. The emission fields for the year 2008 were used.




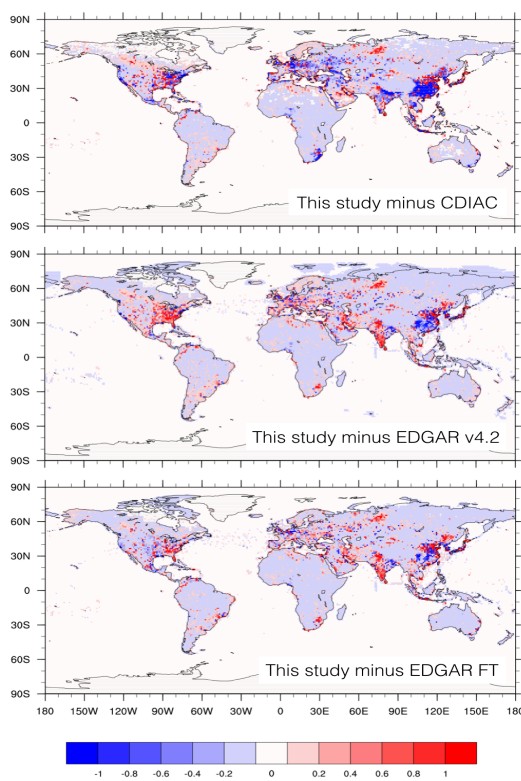

**Figure 7**. ODIAC-other emission data differences. CDIAC (upper right), two versions of EDGAR (v4.2 lower left and v4.2 Fast Track lower right). The units are tonnes carbon/year/cell (1 × 1 degree). Note that the differences are defined as ODIAC (this study) minus others.

6.3 Regional emission time series.
Fig. 8 shows time series of regional fossil fuel emissions aggregated over 11 land regions
defined in the TransCom transport model intercomparison experiment (e.g. Gurney et al.,
2002). The global seasonal variation and the associated uncertainty have been presented and
discussed in Andres et al. (2011). Here monthly total emission values were calculated for
eleven TransCom land regions and presented with the associated uncertainty values (see
Table 3). The monthly total values were calculated in both excluding international bunker
emissions (hence, land emissions only) and including the emissions. The uncertainty range



was calculated by mass weighted uncertainty estimates of countries that fall into the regions. The uncertainty ranges shown in Fig. 8 are annual uncertainty plus the monthly profile uncertainty (12.8%, reported by Andres et al., 2011). Monthly time series are presented for land only emissions and land and international bunker emission (here, largely aviation emissions). As described earlier, the emission seasonality was adopted from Andres et al. (2011). The patterns in emission seasonality are often largely characterized by the large emitting countries within the regions (e.g. U.S. for region 2; China for region 8). Since Andres et al. (2011) used geographical closeness (also, type of economic systems) to define proxy countries, the countries in the same TransCom regions can have similar or the same seasonal patterns in their emissions.

As we can see in Fig. 4 (panel plot for aviation emissions), aviation emissions are intense over North America, Europe and Asia. Global total aviation emission was approximately 0.12 PgC/yr in 2013 and it often does not account for a large portion of the global total (1.2% of the global total in 2013). However, considering the fact that those emissions are concentrated in particular areas such as North America, Europe and East Asia, rather than evenly distributed in space, and often imposed at the surface layer in transport model simulation, care must be taken to achieve an accurate atmospheric $CO_2$ transport model simulations (Nassar et al., 2010). Aviation emissions were often around 0.5-5.1% of the land total emissions over the most regions, but as large as 12.7% (North American Boreal).

## 7. Current limitations, caveats and future prospects

As ODIAC emission data product is now used for a wide variety of carbon cycle research (e.g. global, regional inversions, urban emission studies), it would be useful to note/discuss issues/limitations and caveats in our emission data as well as modeling framework. Some of the issues/limitations are specific to our study, however the majority of them are often shared by existing other gridded emission data and or emission models.




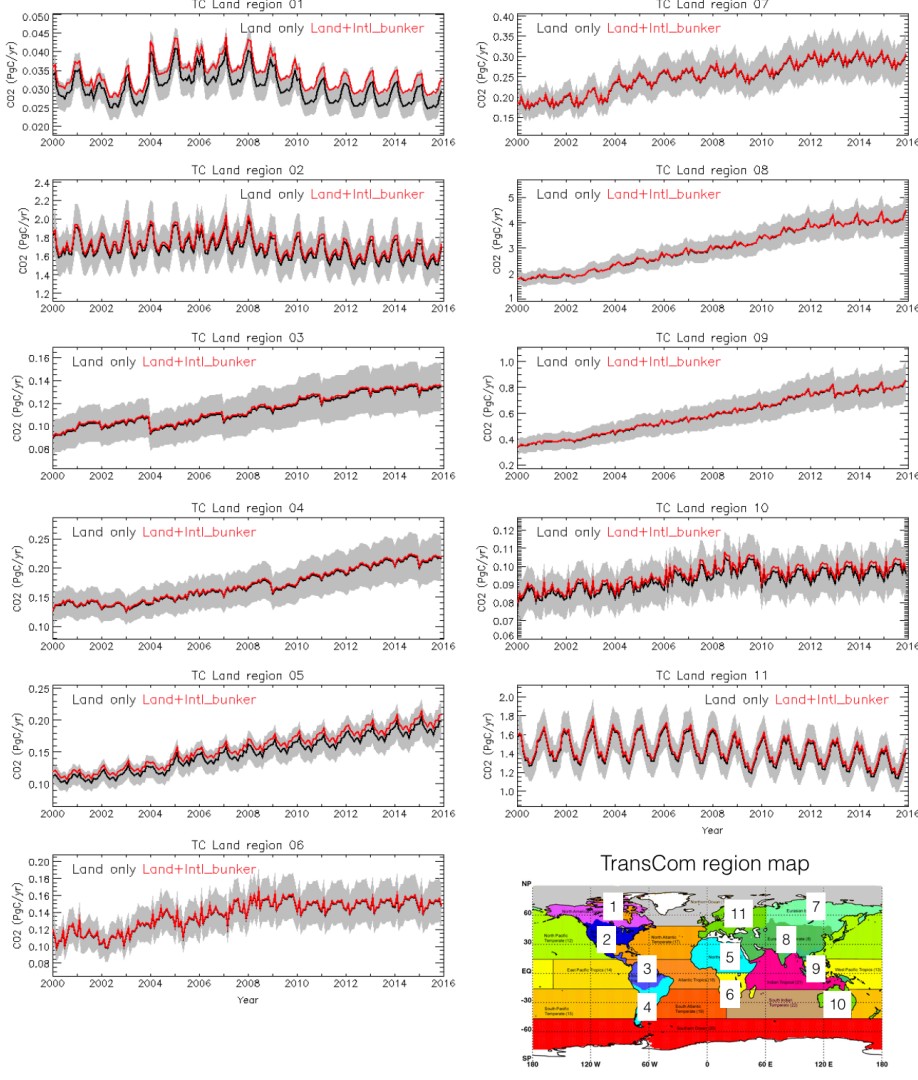

**Figure 8**. Emission time series over inversion analysis land regions defined by the Transport model intercomparison (TransCom) project (Gurney *et al.,* 2002). The TransCom region map (bottom right) is available from http://transcom.project.asu.edu/transcom03_protocol_basisMap.php (last access: 8 November, 2016). Black lines indicate ODIAC 1×1 degree monthly emissions. The monthly emissions are calculated using 1×1 degree ODIAC emission data. The uncertainty range was calculated by mass weighted uncertainty estimates of countries that fall into the regions (see Table 3). The uncertainty ranges shown in Fig. 8 are annual uncertainty plus the monthly profile uncertainty (12.8%, reported by Andres et al., 2011). Note scales in the vertical axis are different.




**Table 3**. Annual total emission over TransCom land regions and the associated uncertainty estimates. The total emissions were calculated using ODIAD2016 gridded emission data. The numbers in the bracket are values including international bunker emissions. The uncertainty estimates were mass weighted values of uncertainty estimates of countries that fall in the regions. Country uncertainty estimates were estimated using the method described Andres et al. (2014). The values were reported as 2-sigma uncertainty.

| Region # | Region name | Uncertainty (%) |
|----------|-------------|-----------------|
| 1 | North American Boreal | 3.7 |
| 2 | North American Temperate | 3.7 |
| 3 | South American Tropical | 9.6 |
| 4 | South American Temperate | 12.8 |
| 5 | Northern Africa | 5.1 |
| 6 | Southern Africa | 10.6 |
| 7 | Eurasian Boreal | 12.4 |
| 8 | Eurasian Temperate | 7.8 |
| 9 | Tropical Asia | 11.8 |
| 10 | Australia | 4.0 |
| 11 | Europe | 3.8 |

7.1 Emission estimates

In the production of ODIAC2016, we used several versions/editions of CDIAC estimates (e.g. global estimates, national estimates and monthly gridded data). This could often happen in emission data production, as some of the underlying data are not updated/upgraded at the time of emission data production (we often start updating emission data after new fuel statistical data are released). We sometimes accept the inconsistency and try to use the most up-to-date information available. For example, we could use GCP's estimates (e.g. Le Quéré et al., 2016) to constrain the global totals, if CDIAC global total emission estimates are not available. The way we obtained emission estimates for each version is often described in netCDF header information of the emission data product. The use of CARMA power plant estimates for estimating magnitude of point source portion of emissions is hard to eliminate, although ideally this is done using emission estimates that are fully compatible to CDIAC estimates. We are currently examining U.N. statistical data (which CDIAC emission estimates are based on) to assess the ability of explaining power plant emissions.

7.2 Emission spatial distributions

7.2.1 Point source emissions

Although the use of the power plant geolocation allowed us to achieve improved high-resolution emission spatial distributions over land (Oda and Maksyutov, 2011), the



availability of power plant data is often very limited. For example, CARMA does not provide
power plant emissions and its status (e.g. commission/decommission) every year and
furthermore update/upgrade after their version 3.0 database (which dated 2012). The error in
their power plant geolocation is another issue that has been identified (e.g. Oda and
Maksyutuov, 2011; Woodard et al., 2015). In ODIAC, the base year emissions (2007) were
projected and all the power plants were assumed to be active over the period (Oda and
Maksyutov, 2011). There are only few global projects that are collecting power plant
information such as the Global Energy Observatory (GEO,
http://globalenergyobservatory.org/) and those can be a useful source of data to improve and
supplement CARMA database. Regionally, CARMA can be evaluated using an inventory
such as the U.S. Emissions and Generation Resource Integrated Database (eGRID) (EPA,
2017). However, it is often difficult to find such a well-constructed and documented
inventory for countries that are actually driving the uncertainty in global emissions (e.g.
China and India).
Emissions from cement production (which are currently distributed using nightlight by
Ziskin et al., 2010) and gas flare (which is distributed using gas flare nightlight data by
Elvidge et al., 2009) should be distributed as point sources. For gas flare emissions, we are
examining the use of Nightfire (Elvidge at al., 2013a) to pinpoint active gas flares in timely
manner and improve their emissions spatial disaggregation over the recent years.
7.2.2 Non-point source emissions
Nighttime light data has been an excellent proxy for human settlements (hence, $CO_2$
emissions) even at a high spatial resolution, however there are some issues to be discussed.
As mentioned earlier, we used an improved version of calibrated radiance data developed by
Ziskin et al. (2010), but those data are only available to seven data periods over the course of
DMSP years (1992-2013). As we do not believe linearly interpolating the existing nightlight
data over the intervening years is necessarily the best way (as done in Asefi-Najafabady et al.,
2014), the same nightlight data has been used for some periods and thus emission
distributions remain unchanged. We are now examining the use of nightlight data collected
from the Visible Infrared Imaging Radiometer Suite (VIIRS) on Suomi National Polar-
orbiting Partnership satellite (e.g. Elvidge et al., 2013b; Román and Stokes, 2015). VIIRS
instruments do not have several critical issues that the DMSP instrument had (e.g. spatial
resolution, dynamic range, quantization and calibration) (Elvidge et al., 2013b). The fully
calibrated nightlight data can be used to map emission changes in space in timely and
consistent manner.
In ODIAC, the disaggregation of non-point emissions is solely done using nighttime light
data for estimating subnational emission spatial distribution and no additional subnational
constrain were applied. Rayner et al. (2010) proposed to better constrain subnational emission
spatial distribution by combining population data, nighttime lights and GDP in their Fossil
Fuel Data Assimilation System (FFDAS) framework. Asefi-Najafabady et al. (2014) further
introduced the use of point source information in their disaggregation, the optimization in
their current framework is however under-constrained by the lack of GDP information.
Without having such optimization, the state level per capita emission estimates can provide
subnational constraints. Nassar et al. (2013) evaluated the per capita emissions in CDIAC and
ODIAC emission data over Canada using the national inventory and found that ODIAC
outperformed. However, as the nightlight-population relationship might have a bias for
developing and the least developed countries (Raupach et al., 2010), we would expect we see



significant biases over those countries and the per capita estimates can provide a useful
constraint.
As seen in the comparison to other emission data, the major difference from EDGAR
emission spatial distribution was due to the lack of line sources in ODIAC. We do not believe
the result from the emission data comparison can falsify the emission distribution in ODIAC,
as discussed earlier. However, we do expect an inclusion of the line sources would improve
the spatial distributions and emission representations in both cities and rural areas. We are
currently examining the inclusion of transportation network data (e.g. OpenStreetMap) as
proxy for line source emissions to explore the better spatial emission aggregation method.
Oda et al. (2017) recently implemented the idea of adding a spatial proxy for line sources and
improved emission estimates for a U.S. city.
7.2.3. Aviation emissions
We estimated emissions from international aviation from CDIAC using U.N. statistical data.
The emissions are currently provided as a single layer emission field, although it is not
appropriate given the nature of the aviation emissions. Nassar et al. (2010) discussed that the
importance of the three dimensional (e.g. x,y,z) emissions for interpreting $CO_2$ profile. In
current modeling framework, although we maintain the aviation emission injection height
from AERO2k (reduced to 1km interval), we distribute the emissions to a single layer. As
pointed out by Olsen et al. (2013), AERO2k does not agree with other inventories in height
distribution. With noting the caution, we will examine the use of height information from
AERO2k and other data available to us and do sensitivity analysis using transport model
simulations.
7.3 Emission temporal profiles.
The emission seasonality in ODIAC2016 is based on Andres et al. (2011) and it can be
further extended using the TIMES scaling parameter to hourly scale. We note that the
emission seasonality was based on top 20 emitting countries' fuel statistics and Monte Carlo
simulation (Andres et al., 2011). The emission seasonality for countries other than the top 20
could be less robust. Also, because of the use of Monte Carlo, the seasonality is different over
different editions of monthly emission data. Andres et al. (2011) estimated the monthly
uncertainty as an additional 12.8% (two sigma) over the annual uncertainty. As we often
impose fossil fuel emissions, care must be taken when applied to inversions. Ultimately, as
done by Vogel et al. (2013), we might be able to evaluate temporal profiles from statistical
data and improve them (but only to limited small locations).
7.4 Uncertainties associated with gridded emission fields
As mentioned earlier, the evaluation of gridded emission data is often very challenging and
most of the emission data study share this difficulty. Although the emission estimates are
made at global and national scales with small uncertainty (e.g. 8% for global by Andres *et al.*,
2014), considerable errors seem to be introduced when disaggregated (e.g. Hogue et al., 2016;
Andres et al., 2016). Andres et al. (2016) for example estimated the uncertainty associated
with CDIAC gridded emission data on a per grid cell basis with an average of 120% and a
range of 4.0 to 190% (2 sigma). Hogue et al. (2016) closely looked at CDIAC gridded



emission data over the U. S. domain and estimated the uncertainty associated with the $1 \times 1$
degree emission grids as ±150%. Those errors seem to be unique to the disaggregation
method (Andres et al., 2016). Future funding may allow us to pursue a full uncertainty
analysis of the ODIAC emission data/model, akin to the Andres et al. (2016) approach but
accounting for the greater than one carbon distribution mechanisms utilized in ODIAC
emission modeling framework. All of the spatially distributed gridded emission data
mentioned in this manuscript suffer from the same basic defect: they use proxies to spatially
distribute emissions rather than actual measurements. In addition, evaluating emission
distributions based on nightlight proxy can be challenging as the connection between $CO_2$
emissions and proxy is less direct compared to population (e.g. per capita emissions). A
combined use of emission proxy and geolocation data (e.g. power plant location) would also
add additional difficulties to give a comprehensive measure of the uncertainty because of
different type of error/uncertainty sources (e.g. Woodard et al., 2015). As finer spatial scales
are approached, the defect of the proxy approach becomes more apparent: proxies only
estimate emission fields. ODIAC data product has been used not only for global simulations
at an aggregated spatial resolution, but also at very high spatial resolution (e.g. Ganshin et al.
2010; Oda et al. 2012; Lauvaux et al. 2016; Oda et al. 2017). Thus, emission evaluation at a
high resolution has become an important task. One approach we could take for evaluating
high-resolution emission fields is comparing to a local fine-grained emission data product
such as Gurney et al. (2012), acknowledging the limitations of the approach discussed earlier.
Another approach would be evaluating emission data in concentration space, rather than
emission space. As reported in Vogel et al. (2013) and Lauvaux et al. (2016), with
radiocarbon measurements and/or good, spatially dense $CO_2$ measurements, a high-resolution
transport model simulation can provide an objective measure for emission data evaluation
(e.g. model-observation mismatch and emission inverse estimate).
While the quality (i.e. bias and uncertainty) of the gridded emission estimates remains
unquantified for most of the emission data mentioned in this manuscript, the emission data
are still used because sufficient measurements in space and time are not presently available to
offer a better alternative. At very least, we presented uncertainty estimates over the
aggregated TransCom land regions. We believe that the regional uncertainty estimates are
highly useful for atmospheric $CO_2$ inversion modelers, more than uncertainty estimates at a
grid level, which still do not seem to be ready for use. Inversion studies often aggregate flux
estimates over the TransCom land regions to interpret regional carbon budgets, while flux
estimations in their models are done at much higher spatial resolutions (e.g. Feng et al., 2009;
Chevallier et al., 2010; Basu et al., 2013). Taking an advantage of being based on CDIAC
estimates, we adopted the updated uncertainty estimates reported by Andres et al. (2016) and
obtained the regional uncertainty estimates. Those estimates are new and readily usable to the
inversion studies especially when interpreting the regional estimates.
**8. Product distribution, data policy and future update**
ODIAC2016 data product is available from a website hosted by the Center for Global
Environmental Research (CGER), Japanese National Institute for Environmental Studies
(NIES) (http://db.cger.nies.go.jp/dataset/ODIAC/, doi: 10.17595/20170411.001). The data
product is distributed under Creative Commons Attribution 4.0 International (CC-BY 4.0,
https://creativecommons.org/licenses/by/4.0/deed.en). ODIAC2016 emission data are
provided in two file formats: 1) global $1\times1$ km (30 arc second) monthly file in GeoTIFF
format (only includes emissions over land) and 2) $1\times1$ degree annual (12 month) file in
netCDF format (includes international bunker emissions). A single, global 1km monthly



GeoTIFF file is about 3.7 GB (compressed to 120 MB). The 1 degree netCDF annual file is about 6MB.

We update the emission data on annual basis, following a release of an updated global fuel statistical data. Future versions of the emissions data are in principle based on updated version/edition of the underlying statistical data with the same name convention (ODIACYYYY, YYYY= the release year, the end year is YYYY minus 1). Currently we are working on the year 2017 version of ODIAC data (ODIAC2017) which covers 2000-2016. We primarily focus on years after 2000. Future versions of ODIAC data however might have a longer, extended time coverage.

**9. Summary**

This manuscript described the year 2016 version of ODIAC emission data (ODIAC2016) and how the emission data product was developed within our upgraded emission modeling framework. Based on CDIAC emission data, ODIAC2016 can be viewed as an extended version of CDIAC gridded data with improved emission spatial distributions representations. Utilizing the best available data (emission estimates and proxy), we achieved a comprehensive, global fossil fuel $CO_2$ gridded emission field that allows data users to impose their $CO_2$ simulations in a consistent way with the global carbon budget analysis. With updated fuel statistics, we should be able to continue producing updated, future versions of ODIAC emission data product within the same model framework. The capability we developed in this study has become more significant now, given CDIAC's shutdown. Despise of expected difficulties (e.g. discontinued CDIAC estimates), the authors believe that ODIAC could play an important role in delivering emission data to the carbon cycle science community. Limitations and caveats discussed in this manuscript mirror and lead ODIAC's future prospects. The ODIAC emission data product is distributed from http://db.cger.nies.go.jp/dataset/ODIAC/ with a DOI. Currently we are working on the 2017 version of ODIAC emission data (ODIAC2017, 2000-2016) and expecting to release by fall 2017.

**Appendix A**

**Table A1**. A list of components in ODIAC2016 and data used in the development.

| Component | Data/product name | Description and data source | Reference |
|---|---|---|---|
| Global FFCO2 | CDIAC global fossil-fuel $CO_2$ emissions | The year 2016 edition of CDIAC global total estimates were used to constrain the ODIAC2016 totals. Data available at http://cdiac.ornl.gov/ ftp/ndp030/global.1751_2013.ems. | Boden et al. (2016) |
| National FFCO2 | CDIAC fossil-fuel $CO_2$ emissions by Nation | The year 2016 editions of CDIAC national emission estimates are used as a primary input data. Data available at http://cdiac.ornl.gov/ ftp/ndp030/nation.1751_2013.ems. | Boden et al. (2016) |
| Global fuel | BP | The year 2016 edition of BP statistical data were used to | BP (2017) |



| statistics | Statistical review of world energy | project CDIAC emissions over the recent years (2014-2015). Data are available at http://www.bp.com/en/global/corporate/energy-economics/statistical-review-of-world-energy.html. | |
|---|---|---|---|
| Monthly temporal variation | CDIAC Gridded Monthly Estimate | The year 2013 version of CDIAC monthy gridded data were used to model seasonality in ODIAC2016. Data are available at http://cdiac.ornl.gov/ftp/fossil_fuel_CO2_emissions_gridded_monthly_v2013/ | Andres et al. (2011) |
| NTL (for non-point emissions) | Global Radiance Calibrated Nighttime Lights | Multiple year NTL data are used to distribute nonpoint emissions. Data are available at https://ngdc.noaa.gov/eog/dmsp/download_radcal.html. | Ziskin et al. (2010) |
| NTL (for gas flaring) | Global Gas Flaring | Global gas flaring NTL data are specifically used to distribute gas flarig emissions. Data are available at http://ngdc.noaa.gov/eog/interest/gas_flares_countries_shapefiles.html | Elvidge et al. (2009) |
| Int'l ship tracks | EDGAR v4.1 | The international marine bunker emission field in EDGAR v4.1 was used. Data are available at http://edgar.jrc.ec.europa.eu/archived_datasets.php. | JRC (2017) |
| Int'l Aviation flight tracks | AERO2k | Data were used to distributed aviation emissions. More details can be find at http://www.cate.mmu.ac.uk/projects/aero2k/. | Eyers et al. (2005) |
| Weekly and diurnal cycle | TIMES | This was not a part of ODIAC2016, however it is useful to note that this scaling factors can be used to create weekly and diurnally varying emissions. Data are available at http://cdiac.ornl.gov/ftp/Nassar_Emissions_Scale_Factors/. | Nasar et al. (2013) |

**Acknowledgments**

TO is supported by NASA Carbon Cycle Science program (Grant # NNX14AM76G). RJA is now retired but this work was sponsored by U.S. Department of Energy, Office of Science, Biological and Environmental Research (BER) programs and performed at Oak Ridge National Laboratory (ORNL) under U.S. Department of Energy contract DE-AC05-00OR22725. The authors would like to thank Chris Elvidge and Kim Baugh at NOAA/NGDC for providing the nightlight data. The authors also thank Yasuhiro Tsukada and Tomoko Shirai for hosting the ODIAC emission data on the data server at NIES.

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
