# Peer review of "The Open-source Data Inventory for Anthropogenic Carbon dioxide (CO2), version"

_Earth System Science Data, 2017_

## Referee Comment (RC1) · Anonymous Referee #1 · 3 Sep 2017

This paper outlines a major update and improvement to the well documented ODIAC emissions inventory data set. With the withdrawal of funding for CDIAC, this data set becomes increasingly important. The fact that this new version makes use of collaborative efforts with CDIAC combines many features of the CDIAC data set that was not formerly a part of the CDIAC emissions inventory is a huge bonus.

Of course, there are some minor issues that need to be cleared up, but they are minor.

[Figure]

In general the presentation is excellent and a few phrasings are particularly nice.

One difficulty is that CDIAC is/was a whole center and has multiple products. In this paper it is sometimes difficult to distinguish between the data products from CDIAC. It would be nice to settle on a standardized way of representing each, perhaps CDIAC_EI for the emissions inventory (?) and something else for the other data.

As the authors know, uncertainty is a big issue and has yet to be incorporated in an appropriate fashion into these inventories. However there is progress and this paper does a better job than most in outlining where its faults lie. There are still some challenges, such as the mysterious global scaling, but I can see ODIAC getting a handle of the uncertainty in the next few years by putting together that various pieces that are already in the literature and cited in this document.

A few technical issues:

1. There are a number of places with missing or incorrect articles ("the", "a"). I think it would be fairly quick for one of the authors to run through the paper with that in mind and do a quick correction. There is not a difficulty with understanding the intent, it would just be a bit smoother.

2. Page 3, line 41. The upgrades to the CDAIC data are never outlined. It might be nice to revisit this at the end to summarize the differences. The following sentence on line 43 needs to be reworded for clarity.

3. Page 4, line 14. "adopt" should be singular.

4. Page 4, line 24. Sentence beginning here is awkward and needs rephrasing.

5. Page 6, line 9-10. This really needs better explanation and resolution. This is kind of a big issue since the world total is simply "scale" to compensate. It sounds like you are doing something "fishy" and rather than actually fixing the issue, that your are just assuming it doesn't matter. It does matter and may be a source, for some countries, of significant error. My suspicion is that a simple scaling may not reflect the proper

distribution of those emissions.

6. on page 7, you outline the use of spatial data of power plants but wait until much later to explain the source of that data. It would be good to reference eGrid here as well. You might also investigate the use of other EPA data products to supplement the eGrid data. If you are planning to eventually pull out concrete production, you might as well get other major industrial sources as well.

7. Several places, such as on page 7, line 38, you explain the use of a data product without citing it. Another place, you cite a paper that also references the data product but not the product itself. The name of the data product should be cited, as well as a paper that might provide an explanation.

8. Page 17, line 25-27. the "/" is awkward and should easily be reworded.

9. The summary has some awkward tense issues. The authors should discuss what should be past tense and what should be present tense. It is this reviewers opinion that anything that the paper does should be present tense and that work done in the past should be past tense.

Overall this is a well written and important paper. Clearing up the few technical issues should be done quickly so that the paper can be officially published.

---

## Referee Comment (RC2) · Anonymous Referee #2 · 12 Sep 2017

Summary: The authors present the new ODIAC2016 gridded (1deg by 1 deg), global, monthly FFCO2 emission dataset and the underlying ODIAC V3.0 emission model. The ODIAC model relies on multiple open-source datasets to improve the existing emission inventory provided by CDIAC. The specific target is to provide a better spatial and temporal disaggregation of sources (using e.g. nightlights, gas flaring, bunker fuel statistics), which is a critical improvement, if this dataset is to be used in atmospheric transport models or large scale inversion systems. The authors discuss the construc-

tion of the new emission model and its limitations. The study is well-written and the scientific methods chosen are sound. Despite the large amount of processes captured, the paper is quite comprehensive in its description, but should focus more on one critical point its discussion – the vertical disaggregation of emissions in ODIAC v3. This will be highly relevant for regional scale atmospheric transport modellers, but can be easily added to the existing manuscript. If this general comment and minor comments are addressed, I fully recommend the publication of the manuscript and would assume that this dataset is of extraordinary interest to the community of regional and global GHG modellers as it is unique in its approach and of high quality. General comments: The section on spatial disaggregation fails to clearly address the issue of emission heights. Aviation emissions are distributed according to AERO2k but are then aggregated to a single layer – but at which height? Especially, for Northern Canada, where emissions increase by 5 orders of magnitude in some regions in ODIAC2016 compared to CDIAC the chosen injection height might be critical. Furthermore, it is unclear which emission height is assumed for flares and point sources – the impact on regional scale models could be significant. This should be discussed more clearly. Specific comments: Caption figure 1:'ODAIC 3.0' -> 'ODIAC 3.0' P3 L43: please be more specific – 'timely manner' = we can expect annual release of updates to ODIAC2016? P6 L17: 'carbo' -> carbon P7 L11: The product discussed in this manuscript has 1deg by 1deg resolution according to P4L32, but section 4 is not always explicit about the resolution of the disaggregation (or this is sector specific). I assume 1km by 1km was used for some sectors and then a re-aggregation to 1deg by 1deg was performed for the global ODIAC2016 product? P7 L13: Please specify at which resolution the data was disaggregated here P7 L18: Needs details on what is considered a point source (only data from a specific database or a emission rate per site?) and what is a non-point source. P7L29 indicates that non-point source is the default category for point sources that cannot be correctly located. P7L36: What happens to other emissions in flare pixels? What is the impact of overlaps of urbanised area nightlights and O&G extraction regions e.g. in the Barnett shale (Dallas-Fort Worth region) or Niger Delta? Is this a potential bias or

insignificant? P8L10: 'we does not' -> 'we do not' P8L24: was this climatology based on external driver (e.g. correlation of seasonal emission changes with seasonal temperature changes/anomalies) or just mean seasonal cycles of emissions for the years 2000-2010? A recent study by Breon et al. suggests a significant impact of temperature anomalies on FFCO2 emissions (http://iopscience.iop.org/article/10.1088/1748-9326/aa693d) and we could expect an impact on seasonality of emissions from this as well. Fig2. Refers to CDIAC/GLB+projection while Table 1. Ignores this and shows ODIAC2016 in comparison to the three others. Please consider removing this inconsistency. Fig3. Caption: 'top 20 emitting' -> top 10 emitting Figure 6 and 7: I assume the caption or colour scale is wrong here and units are NOT 'The units are tonnes carbon/year/ cell (1×1 degree)'

Emissions of 1 tonne carbon/(a*cell) a 1deg by 1deg grid cell seem extremely unreasonable. Additional suggestion: Show a histogram of differences between ODIAC and other inventories here (or in appendix).

---

## Author Comment (AC1) · 20 Nov 2017

Dear Anonymous Referee #2

*Summary: The authors present the new ODIAC2016 gridded (1deg by 1 deg), global, monthly FFCO2 emission dataset and the underlying ODIAC V3.0 emission model. The ODIAC model relies on multiple open-source datasets to improve the existing emission inventory provided by CDIAC. The specific target is to provide a better spatial and temporal disaggregation of sources (using e.g. nightlights, gas flaring, bunker fuel statistics), which is a critical improvement, if this dataset is to be used in atmospheric transport models or large scale inversion systems. The authors discuss the construction of the new emission model and its limitations. The study is well-written and the scientific methods chosen are sound. Despite the large amount of processes captured, the paper is quite comprehensive in its description, but should focus more on one critical point its discussion – the vertical disaggregation of emissions in ODIAC v3. This will be highly relevant for regional scale atmospheric transport modellers, but can be easily added to the existing manuscript. If this general comment and minor comments are addressed, I fully recommend the publication of the manuscript and would assume that this dataset is of extraordinary interest to the community of regional and global GHG modellers as it is unique in its approach and of high quality.*

Thank you for your review and the time.  I fully agree that the vertical disaggregation is very important from atmospheric modeling viewpoint.  See our response to your comments below.

*General comments: The section on spatial disaggregation fails to clearly address the issue of emission heights. Aviation emissions are distributed according to AERO2k but are then aggregated to a single layer – but at which height? Especially, for Northern Canada, where emissions increase by 5 orders of magnitude in some regions in ODIAC2016 compared to CDIAC the chosen injection height might be critical. Furthermore, it is unclear which emission height is assumed for flares and point sources – the impact on regional scale models could be significant. This should be discussed more clearly.*

We fully agree with the reviewer that on the emission injection height information is important.  As the reviewer pointed out, aviation emissions are concentrated over places like North America, Europe and East Asia and the representation of those emissions should have some impact especially in regional simulations.  In ODIAC model, we internally carry an injection height information from AERO2k.  We however provide the aviation emissions as a part of international bunker field as a single layer, without specifying certain height.  Providing the emission in a single layer is a common practice as done by other existing gridded emission inventories such as EDGAR (and FFDAS which adopts EDGAR international bunker emissions as is).  We are currently studying the sensitivity study of the injection height using transport models, as discussed in 7.2.3.  Upon completion of the sensitivity study, we would like to document and report in a future manuscript and include the injection height information in ODIAC emission product.

We also do not have injection height information for other emission sources such as power plants and gas flares. This is simply because of lack of the global data. We believe many of other global emission data products share this difficulty. We propose to add this sentence at the end of 7.2.1.

"Currently, the point source emissions in ODIAC do not have an injection height due to the lack of global information. This limitation is share with other existing global emission data products."

Specific comments:

*Caption figure 1: 'ODAIC 3.0' -> 'ODIAC 3.0'*

Fixed.

*P3 L43: please be more specific – 'timely manner' = we can expect annual release of updates to ODIAC2016?*

Yes, you are correct. With the phrase "timely manner", we wanted to emphasize that we plan to work on annual emission data update as soon as underlying data become available and deliver the data to the science community. We propose to rephrase the sentence

"Our emission modeling framework was also designed to produce an emission data product in a timely manner, with updated information."

As

"Our emission modeling framework was also designed to produce an annually-updated emission data product in a timely manner.

*P6 L17: 'carbo' -> carbon*

Fixed.

*P7 L11: The product discussed in this manuscript has 1deg by 1deg resolution according to P4L32, but section 4 is not always explicit about the resolution of the disaggregation (or this is sector specific). I assume 1km by 1km was used for some sectors and then a re-aggregation to 1deg by 1deg was performed for the global ODIAC2016 product?*

You are correct. We propose to indicate spatial resolution (1km) of disaggregation in the main text where it is missing. Most of the places we believe we indicate the emission disaggregation spatial resolution. We hesitate to add the spatial resolution in the section title as we also provide the 1-deg product. We also propose to add the word "aggregated" for the 1-deg product as appropriate for

clarification (See Section 2).  Please see the revised manuscript.

*P7 L13: Please specify at which resolution the data was disaggregated here*

See our response above.

*P7 L18: Needs details on what is considered a point source (only data from a specific database or a emission rate per site?) and what is a non-point source. P7 L29 indicates that non-point source is the default category for point sources that can- not be correctly located.*

We used CARMA database (emission intensity and geolocation) to map point source emissions.  Unlike EDGAR database, CDIAC emission data are fuel-based emission estimates (similar to IPCC reference method) and do not distinguish specific IPCC-like sector emissions.  We have been used CARMA to divide land emissions into two emission types (namely point and non-point).  We dedicated 3.3 to describe how we re-categorize CDIAC emission categories to our own ODIAC categories.

The sentence at P7 P29 mentioned that cement production emissions are ideally mapped as a point source.  But simply due to the lack of global information, we distributed the cement emissions as a non-point source using nightlight data.  As described in section 3.3 and after, the nonpoint source category is not for the emissions that cannot be correctly located.  I believe this was clear with the sentence.

*P7 L36: What happens to other emissions in flare pixels? What is the impact of overlaps of urbanised area nightlights and O&G extraction regions e.g. in the Barnett shale (Dallas-Fort Worth region) or Niger Delta? Is this a potential bias or insignificant?*

Although we map those emissions separately w/o overlapping, we are using the separate gas flaring nightlight data to map the emissions, but the nightlight data do not include recent emission changes.  This is a potential source of biases due to our model representation error.  As we discussed in the manuscript, we hope the use of VIIRS data would at least mitigate the errors.  We expect the use of VIIRS data would allow us to reduce the representation and mapping errors and quantify the biases.

*P8L10: 'we does not' -> 'we do not'*

Fixed.

*P8L24: was this climatology based on external driver (e.g. correlation of seasonal emission changes with seasonal temperature changes/anomalies) or just mean seasonal cycles of emissions for the years 2000-2010? A recent study by Breon et al. suggests a significant impact of temperature anomalies on FFCO2 emissions (http://iopscience.iop.org/article/10.1088/1748- 9326/aa693d) and we could expect an impact on seasonality of emissions from this as well.*

The climatology is the mean seasonal cycles of national emissions.  Thank you for referring to Breon et al. (2017) work which we were not aware.  The use of HDD/CDD is a common approach to infer seasonal changes in energy-related emissions.  We should be able to do some correction to the mean or create seasonality, but it is uncertain if that would make our seasonal estimates close to the truth.  It would make sense to do logically, but there is no objective measure to confirm the expected improvement.  It is possible that we might add some biases.  We prefer to stick to CDIAC seasonality that is developed based on statistical data.  We acknowledged the potential biases at 7.3 in the manuscript

*Fig2. Refers to CDIAC/GLB+projection while Table 1. Ignores this and shows ODIAC2016 in comparison to the three others. Please consider removing this inconsistency.*

CDIAC/GLB+proj and total ODIAC2016 are identical.  We propose to change the legend in Figure 2

from "CDIAC/GLB+Projected"

to

 "CDIAC/GLB+Projected (ODIAC2016)".

*Fig3. Caption: 'top 20 emitting' -> top 10 emitting*

Fixed.

*Figure 6 and 7: I assume the caption or colour scale is wrong here and units are NOT 'The units are tonnes carbon/year/ cell (1×1 degree)'  Emissions of 1 tonne carbon/(a\*cell) a 1deg by 1deg grid cell seem extremely unreasonable. Additional suggestion: Show a histogram of differences between ODIAC and other inventories here (or in appendix).*

Thank you so much for catching this.  The unit is million tonnes of carbon per year (MTC/yr).  "Unit: MTC/yr" will be to Figure 6 and 7. We created a set of histograms based on the same result.  We plan to add this figure as Appendix A2.

[Figure]

Fig. A3. Histogram of the inter-emission data differences from ODIAC. Values are given in the unit of million tonnes carbon per year (MTC/yr).

In addition to the reviewer's suggestions, we propose to add some text to describe the updated year 2017 versions of the ODIAC emission data product (ODIAC2017, 2000-2017). We also made minor editorial modifications to the main text to improve the readability. See the revised manuscript.

Thank you so much for your comments and suggestions.

---

## Author Comment (AC2) · 20 Nov 2017

Dear Anonymous Referee #1

First of all, we appreciate your comments and suggestions. Our responses are inlined.

*This paper outlines a major update and improvement to the well documented ODIAC emissions inventory data set. With the withdrawal of funding for CDIAC, this data set becomes increasingly important. The fact that this new version makes use of collaborative efforts with CDIAC combines many features of the CDIAC data set that was not formerly a part of the CDIAC emissions inventory is a huge bonus. Of course, there are some minor issues that need to be cleared up, but they are minor. In general the presentation is excellent and a few phrasings are particularly nice.*

Thank you for your comment and time you spent for this review.

*One difficulty is that CDIAC is/was a whole center and has multiple products. In this paper it is sometimes difficult to distinguish between the data products from CDIAC. It would be nice to settle on a standardized way of representing each, perhaps CDIAC_EI for the emissions inventory (?) and something else for the other data.*

It is a great idea to introduce a standardized way of addressing CDIAC and its research products. We have tried, but so far we have not been able to come up with a reasonable solution for this. At very least, to avoid the confusion, we propose to specifically add words such as "gridded" or "emission estimates", and "global", "national" or "monthly" where appropriate. We address CDIAC (lab) as CDIAC/ORNL. We hope this would improve the readability.

*As the authors know, uncertainty is a big issue and has yet to be incorporated in an appropriate fashion into these inventories. However there is progress and this paper does a better job than most in outlining where its faults lie. There are still some challenges, such as the mysterious global scaling, but I can see ODIAC getting a handle of the uncertainty in the next few years by putting together that various pieces that are already in the literature and cited in this document.*

As we discussed in the manuscript, the evaluation of uncertainty in gridded emission data products are challenging primarily because of the lack of physical measurements, especially at an aggregated spatial resolution. As you mentioned, with new data and modeling capability, we are trying to work on uncertainties with a hope of distributing future ODIAC emission products with reasonable uncertainty estimates. We expect the use of VIIRS nightlight data, as described in the manuscript, also will allow us to conduct a rigorous uncertainty analysis. Although it would only address an uncertainty that is specific to ODIAC model structure, we believe it would be a tiny, but significant progress from the current DMSP-based ODIAC. Later in this response, we will discuss the global scaling we implemented in ODIAC2016.

A few technical issues:

*1. There are a number of places with missing or incorrect articles ("the", "a"). I think it would be fairly quick for one of the authors to run through the paper with that in mind and do a quick correction. There is not a difficulty with understanding the intent, it would just be a bit smoother.*

Thank you for your suggestion.  Please see the revised manuscript.  Hope we improve the readability.

*2. Page 3, line 41. The upgrades to the CDAIC data are never outlined. It might be nice to revisit this at the end to summarize the differences. The following sentence on line 43 needs to be reworded for clarity.*

We propose to remove the word "upgrade", as the word "extend" is more appropriate to describe our use of CDIAC data (e.g. the combined use of global and national emission estimates, emission seasonality and international bunker).

We propose to rephrase the sentence on L43

 "As our ODIAC data product is based CDIAC emission data, our emission data production capability is significant given the expected discontinuity of future CDIAC emission data. "
as

"Given the expected discontinuity of future, updated CDIAC emission data, we believe our emission data production capability of producing an extended product of the CDIAC emission data is significant."

*3. Page 4, line 14. "adopt" should be singular.*

Fixed.

*4. Page 4, line 24. Sentence beginning here is awkward and needs rephrasing.*

We propose to rephrased the sentence at Page 4, Line 24

"Given recent most of atmospheric $CO_2$ inversion studies focused on years after 2000, we put a priority to develop emission data for years after 2000 and deliver to the science community in a timely manner."

as

"Atmospheric $CO_2$ inversion studies recently published (e.g. Maksyutov et al. 2013)and operational assimilation systems such as NOAA's CarbonTracker (https://www.esrl.noaa.gov/gmd/ccgg/carbontracker/) often focus on time periods after 2000.  We thus put a priority to produce emission data after year 2000 with regular update upon the availability of updated emission and fuel statistical data and deliver the emission product to the science community,

instead of developing a longer term emission data product."

*5. Page 6, line 9-10. This really needs better explanation and resolution. This is kind of a big issue since the world total is simply "scale" to compensate. It sounds like you are doing something "fishy" and rather than actually fixing the issue, that your are just assuming it doesn't matter. It does matter and may be a source, for some countries, of significant error. My suspicion is that a simple scaling may not reflect the proper distribution of those emissions.*

We would like to make it clear that we do not claim that applying a scaling factor is an ultimate solution to reconcile two different estimates.  As described in the manuscript, global emissions and the sum of national totals do not match because of the differences in the emission estimation methodologies (e.g. Andres et al. 2012), although theoretically/conceptually those two should match.  Those two are independent estimates.  The difference is not due to something like errors in calculations and unlikely to go away completely.  We consider the difference is a manifestation of the resolution/precision of global emission estimates, rather than an issue in emission calculation.  Defining the difference as an issue and seek an ultimate solution is out of scope of this particular manuscript, although it is very important.

What we think more problematic is that different, inconsistent fossil fuel emissions are used in the scientific community to infer carbon fluxes (for example, GCP and atmospheric inversion).  The global scaling is not physical, but we still decided to do it as we believe it should be useful for atmospheric inversion perspective because data users can impose their inverse models with the same global fossil fuel emission estimates (= add the same about of fossil fuel C as other carbon budget studies like GCP, to their models).  The difference is mainly due to the inconsistency in underlying statistical data which seems to be nearly impossible to reduce to zero, although some significant effort of improving statistical data collection system might reduce the difference.  We also did not claim the 3% correction does not matter.  Given the size of the national total uncertainties (4-20% at national level according to Andres et al. 2014), the scaling adjustment can be done within the uncertainty range.  We decided to stick to the global total by sacrificing the accuracy of national emission estimates.  The simple global scaling is also easily removed.  To us, the few percent is also the uncertainty we need to accept in the analysis of carbo budget analyst (representation error).  We propose to add a table for the correction factor we used. Then users can remove the correction if they prefer.  We would like to see how many people would remove the scaling factor from the emissions.

Table A2. A table for the global scaling factor for 2000-2013.

| Year | Scaling factor |
|------|----------------|
| 2000 | 0.999 |
| 2001 | 1.016 |
| 2002 | 1.008 |

| | |
|---|---|
| 2003 | 1.014 |
| 2004 | 1.012 |
| 2005 | 1.022 |
| 2006 | 1.022 |
| 2007 | 1.016 |
| 2008 | 1.023 |
| 2009 | 1.024 |
| 2010 | 1.015 |
| 2011 | 1.017 |
| 2012 | 1.017 |
| 2013 | 1.025 |

*6. on page 7, you outline the use of spatial data of power plants but wait until much later to explain the source of that data. It would be good to reference eGrid here as well. You might also investigate the use of other EPA data products to supplement the eGrid data. If you are planning to eventually pull out concrete production, you might as well get other major industrial sources as well.*

Thank you for catching this.  Our power plant information in ODIAC is primarily based on CARMA, which are partially based on eGRID.  We would like to add a reference for CARMA (Wheeler and Ummel, 2008) to the main text also to the Appendix A.  We are also working on an EPA data-based emission map and plan to include some of the outcome from the study to future versions of ODIAC.  We would like to document the effort in a separate, future manuscript.  Thank you for your suggestion for the cement production emissions.  The future plan was briefly discussed in the uncertainty section.

*7. Several places, such as on page 7, line 38, you explain the use of a data product without citing it. Another place, you cite a paper that also references the data product but not the product itself. The name of the data product should be cited, as well as a paper that might provide an explanation.*

We added citations as suggested.  Please see the revised manuscript. As mentioned at P4, L19-21, all the data used in the ODIAC emission development and associated citations and data sources are also summarized in Appendix A.

*8. Page 17, line 25-27. the "/" is awkward and should easily be reworded.*

We propose to fix the sentence

"…it would be useful to note/discuss issues/limitations and caveats in our emission data as well as modeling framework. Some of the issues/limitations are specific to our study, however the majority of them are often shared by existing other gridded emission data and or emission models."

as

"… it would be useful for the users of the ODIAC emission data product to note and discuss issues, limitations and caveats in our emission data that the authors aware.  Some of the issues and limitations are specific to our study, however the majority of them are shared by other existing gridded emission data and emission models."

9. The summary has some awkward tense issues. The authors should discuss what should be past tense and what should be present tense. It is this reviewers opinion that anything that the paper does should be present tense and that work done in the past should be past tense.

Thank you for your suggestion.  Please see the revised manuscript.  Hope we improve the readability.

*Overall this is a well written and important paper. Clearing up the few technical issues should be done quickly so that the paper can be officially published.*

In addition to the reviewer's suggestions, we propose to add some text to describe the updated year 2017 versions of the ODIAC emission data product (ODIAC2017, 2000-2017).  We also made minor editorial modifications to the main text to improve the readability.  Please see the revised manuscript.

Thank you so much for your comments and suggestions.

---

## Author Comment (AC3) · 20 Nov 2017

[revised manuscript text omitted]
 (2011). Currently the updated, year 2017 version of the ODAIC emission data (ODIAC2017, 2000-2016) are available. This manuscript however provides the sufficient details of how we developed ODIAC2017 with updated information.

**2. Emission modeling framework**

   Fig. 1 illustrates our current ODIAC emission modeling framework (we defined it as "ODIAC 3.0 model", in contrast to the original version). Major changes/differences from Oda and Maksyutov (2011, ODIAC v1.7) are 1) the use of emissions estimates made by the CDIAC/ORNL (rather than our own emission estimates), 2) the use of multiple spatial emission proxies in order to distribute CDIAC national emissions estimates made by fuel type, and 3) the inclusion of emission temporal variations (version 1.7 only indicates annual emission fields). Given CDIAC emission estimates have been one of well-respected, widely-used in the carbon research community (e.g. Ballantyne et al., 2012; Le Quéré et al., 2016), our philosophy in our emission data development is we develop and deliver an extended, comprehensive global gridded emission data product, fully utilizing CDIAC emissions data (e.g. emission estimates in both tabular and gridded forms). We also extend CDIAC emission data where possible. Our emission modeling framework was also designed to produce an annually-updated emission data product in a timely manner. Given the discontinuity of future, updated CDIAC emission data, we believe our emission data production capability of producing an extended product of the CDIAC emission data is significant.

Starting with national emission estimates as an input, our model framework achieves monthly, global FFCO2 gridded fields via preprocessing, and spatial and temporal disaggregation. CDIAC national estimates made by fuel type (liquid, gas, solid, cement production, gas flare and international bunker emissions) are further divided into an extended set of ODIAC emission categories (point source, non-point source, cement production, gas flare, international aviation and marine bunker (further described in Section 3). It is important to note that ODIAC2016 carries emissions from international bunker (international marine bunker and aviation, often accounts for a few percent of the global total emissions), which are not included in the CDIAC gridded emission data products (CDIAC gridded emission data only indicate national emissions and international bunker emissions are often not considered to be a part of national emissions in an international convention). With the inclusion of international bunker emissions, we provide a more comprehensive global gridded emission field. We extended the CDIAC national estimates over the recent years that was not yet covered in the version of CDIAC gridded data (2014-2016), in order to support near-real time $CO_2$ simulations/analysis. Emissions are then spatially distributed using a wide variety of spatial data (e.g. point source geographical location, nighttime light data and flight/ship tracks, further described in Section 4). We adopt an emission seasonality from existing emission inventories for particular emission categories (further described in Section 5).

In the following sections (Section 3-5), we describe how ODIAC2016 was developed. It is important to note that ODIAC2016 is based on the best available data at the time of the development (ODIAC2016 was released in September 2016). Thus, some of the emission estimates and underlying data used in ODIAC2016 might have been outdated. For traceability purpose, data used in this development, their versions/editions, and data sources are summarized in Appendix A. Following the results and evaluation section (Section 6), we discuss caveats and current limitations in our modeling framework/emission data product (Section 7), and then describe how we will update the ODIAC emission data product with updated fuel statistics and/or emission information (Section 8). Atmospheric $CO_2$ inversion studies recently published (e.g. Maksyutov et al., 2013) and operational assimilation systems such as NOAA's CarbonTracker (https://www.esrl.noaa.gov/gmd/ccgg/carbontracker/) often focus on time periods after 2000. We thus put a priority to produce emission data after year 2000 with regular update upon the availability of updated emission and fuel statistical data and deliver the emission product to the science community, instead of developing a longer term emission data product. Future versions of ODIAC data however might have a longer, extended time coverage. Currently the ODIAC data are provided in two data formats: 1) global 1×1 km (30 arc second) monthly data in the GeoTIFF format (only includes emissions over land) and 2) 1×1 degree annual (12 month) data in the NetCDF format (includes international bunker emissions). The 1×1 degree annual data are aggregated from the 1×1 km product. 
[revised manuscript text omitted]

the point source emissions in ODIAC do not have an injection height due to the lack of global
information. This limitation is shared with other existing global emission data products.
7.2.2 Non-point source emissions
Nighttime light data has been an excellent proxy for human settlements (hence, $CO_2$
emissions) even at a high spatial resolution, however there are some issues to be discussed.
As mentioned earlier, we used an improved version of calibrated radiance data developed by
Ziskin et al. (2010), but those data are only available to seven data periods over the course of
DMSP years (1992-2013). As we do not believe linearly interpolating the existing nightlight
data over the intervening years is necessarily best way (as done in Asefi-Najafabady et al.,
2014), the same nightlight data has been used for some periods and thus emission
distributions remain unchanged. We are now examining the use of nightlight data collected
from the Visible Infrared Imaging Radiometer Suite (VIIRS) on Suomi National Polar-
orbiting Partnership satellite (e.g. Elvidge et al., 2013b; Román and Stokes, 2015). VIIRS
instruments do not have several critical issues that the DMSP instrument had (e.g. spatial
resolution, dynamic range, quantization and calibration) (Elvidge et al., 2013b). The fully
calibrated nightlight data can be used to map emission changes in space in timely and
consistent manner.
In ODIAC, the disaggregation of non-point emissions is solely done using nighttime light
data for estimating subnational emission spatial distributions and no additional subnational
emission constrain were applied. Rayner et al. (2010) proposed to better constrain subnational
emission spatial distribution by combining population data, nighttime lights and GDP in their
Fossil Fuel Data Assimilation System (FFDAS) framework. Asefi-Najafabady et al. (2014)
further introduced the use of point source information in their disaggregation, the
optimization in their current framework is however under-constrained by the lack of GDP
information. Without having such optimization, the state level per capita emission estimates
can provide subnational constraints. Nassar et al. (2013) evaluated the per capita emissions in
CDIAC and ODIAC emission data over Canada using the national inventory and found that

ODIAC outperformed. However, as the nightlight-population relationship might have a bias for developing and the least developed countries (Raupach et al., 2010), we would expect we see significant biases over those countries and the per capita estimates can provide a useful constraint.

As seen in the comparison to other emission data, the major difference from EDGAR emission spatial distribution was due to the lack of line sources in ODIAC. We do not believe the result from the emission data comparison can falsify the emission distribution in ODIAC, as discussed earlier. However, we do expect an inclusion of the line sources would improve the spatial distributions and emission representations in both cities and rural areas. We are currently examining the inclusion of transportation network data (e.g. OpenStreetMap) as proxy for line source emissions to explore the better spatial emission aggregation method. Oda et al. (2017) recently implemented the idea of adding a spatial proxy for line sources and improved emission estimates for a U.S. city.

7.2.3. Aviation emissions

We estimated emissions from international aviation from CDIAC using U.N. statistical data. The emissions are currently provided as a single layer emission field, although it is not appropriate given the nature of the aviation emissions. Nassar et al. (2010) discussed that the importance of the three dimensional (e.g. x,y,z) emissions for interpreting $CO_2$ profile. In current modeling framework, although we maintain the aviation emission injection height from AERO2k (reduced to 1km interval), we distribute the emissions to a single layer. As pointed out by Olsen et al. (2013), AERO2k does not agree with other inventories in height distribution. With noting the caution, we will examine the use of height information from AERO2k and other data available to us and do sensitivity analysis using transport model simulations.

7.3 Emission temporal profiles.

The emission seasonality in ODIAC2016 is based on Andres et al. (2011) and it can be further extended using the TIMES scaling parameter to hourly scale. We note that the emission seasonality was based on top 10 emitting countries' fuel statistics and Monte Carlo simulation (Andres et al., 2011). The emission seasonality for countries other than the top 10 could be less robust. Also, because of the use of Monte Carlo, the seasonality is different over different editions of monthly emission data. It is also important to note that the repeated use of climatological (mean) seasonality for the recent years (described in Section 5) could be a source of uncertainty and biases. 
[revised manuscript text omitted]
 | CDIAC | The year 2016 editions of the CDIAC national emission | Boden et al. |

| | | | |
|---|---|---|---|
| FFCO2 | fossil-fuel $CO_2$ emissions by Nation | estimates are used as a primary input data. Data available at http://cdiac.ornl.gov/ ftp/ndp030/nation.1751_2013.ems. | (2016) |
| Global fuel statistics | BP Statistical review of world energy | The year 2016 edition of the BP statistical data were used to project CDIAC national emissions over the recent years (2014-2015). Data are available at http://www.bp.com/en/global/corporate/energy-economics/statistical-review-of-world-energy.html. | BP (2017) |
| Monthly temporal variation | CDIAC Gridded Monthly Estimate | The year 2013 version of the CDIAC monthly gridded data were used to the model seasonality in ODIAC2016. Data are available at http://cdiac.ornl.gov/ ftp/fossil_fuel_CO2_emissions_gridded_monthly_v2013/ | Andres et al. (2011) |
| Power plant data | CARMA | The CARMA power plant database with geolocation correction described in Oda and Maksyutov (2011). Data available from http://carma.org/. | Wheeler and Ummel et al. 2008 |
| NTL (for non-point emissions) | Global Radiance Calibrated Nighttime Lights | Multiple year NTL data are used to distribute nonpoint emissions. Data are available at https://ngdc.noaa.gov/eog/dmsp/download_radcal.html. | Ziskin et al. (2010) |
| NTL (for gas flaring) | Global Gas Flaring Shapefiles | Global gas flaring NTL data are specifically used to distribute gas flaring emissions. Data are available at http://ngdc.noaa.gov/eog/interest/ gas_flares_countries_shapefiles.html | Elvidge et al. (2009) |
| Int'l ship tracks | EDGAR v4.1 | The international marine bunker emission field in EDGAR v4.1 was used. Data are available at http://edgar.jrc.ec.europa.eu/archived_datasets.php. | JRC (2017) |
| Int'l Aviation flight tracks | AERO2k | Data were used to distributed aviation emissions. More details can be find at http://www.cate.mmu.ac.uk/projects/aero2k/. | Eyers et al. (2005) |
| Weekly and diurnal cycle | TIMES | This was not a part of ODIAC2016, however it is useful to note that this scaling factors can be used to create weekly and diurnally varying emissions. Data are available at http://cdiac.ornl.gov/ftp/Nassar_Emissions_Scale_Factors/. | Nasar et al. (2013) |

**Appendix A2**

**Table A2.** A table for the global scaling factor for 2000-2013.

| Year | Scaling factor |
|---|---|
| 2000 | 0.999 |
| 2001 | 1.016 |
| 2002 | 1.008 |
| 2003 | 1.014 |
| 2004 | 1.012 |
| 2005 | 1.022 |
| 2006 | 1.022 |

| | |
|---|---|
| 2007 | 1.016 |
| 2008 | 1.023 |
| 2009 | 1.024 |
| 2010 | 1.015 |
| 2011 | 1.017 |
| 2012 | 1.017 |
| 2013 | 1.025 |

**Appendix A3**

Fig. A3. A histogram of the inter-emission data differences from ODIAC. Values are given in the unit of million tonnes carbon per year (MTC/yr).

[revised manuscript text omitted]